# Deep-Q-Network-Based Packet Scheduling in an IoT Environment

**DOI:** 10.3390/s23031339

**Published:** 2023-01-25

**Authors:** Xing Fu, Jeong Geun Kim

**Affiliations:** Department of Electrical Engineering, College of Electronics and Information, Kyung Hee University, Yongin 17104, Republic of Korea

**Keywords:** energy efficiency, deep Q-network, reinforcement learning, IoT, wireless sensor network, BLE

## Abstract

With the advent of the Internet of Things (IoT) era, a wide array of wireless sensors supporting the IoT have proliferated. As key elements for enabling the IoT, wireless sensor nodes require minimal energy consumption and low device complexity. In particular, energy-efficient resource scheduling is critical in maintaining a network of wireless sensor nodes, since the energy-intensive processing of wireless sensor nodes and their interactions is too complicated to control. In this study, we present a practical deep Q-network (DQN)-based packet scheduling algorithm that coordinates the transmissions of multiple IoT devices. The scheduling algorithm dynamically adjusts the connection interval (CI) and the number of packets transmitted by each node within the interval. Through various experiments, we verify how effectively the proposed scheduler improves energy efficiency and handles the time-varying nature of the network environment. Moreover, we attempt to gain insight into the optimized packet scheduler by analyzing the policy of the DQN scheduler. The experimental results show that the proposed scheduling algorithm can further prolong a network’s lifetime in a dynamic network environment in comparison with that in other alternative schemes while ensuring the quality of service (QoS).

## 1. Introduction

The IoT has brought unprecedented convenience in everyday life and triggered various technological innovations throughout industries [1,2,3]. The core technological elements enabling the IoT include wireless sensor networks (WSNs), cloud computing, data analytics, smart devices, etc. [1,2]. A WSN is responsible for transferring the information obtained from sensor nodes to remote sites [4,5]. The sensor nodes constituting a WSN often rely on batteries or energy harvesting as an energy source [6]. Regardless of the type of energy source that is in use, improving energy efficiency remains a critical issue, since it determines the quality of network connectivity [7,8].

There have been numerous attempts to enhance the energy efficiency of WSNs by employing various technical methods [3,4,5,6]. In particular, energy efficiency can be further improved by utilizing emerging approaches, such as artificial intelligence (AI)-based algorithms [9,10,11,12]. In [9], Bhandari et al. introduced a support vector machine (SVM)-based packet scheduling method for wireless networks. The scheduler considered the channel capacity and average throughput to determine the user–channel mapping. A deep neural network (DNN) was utilized by Zhang et al. in [10] for a channel allocation algorithm in interference-limited wireless networks. The DNN therein approximated a complex function of a traditional sequential convex approximation (SCA)-based algorithm, thereby significantly reducing the complexity of the allocation algorithm. In [11], Xu et al. proposed DQN-based optimal link scheduling in a dense cellular environment. The DQN therein gave an estimated schedule of link and user selection, and then power is allocated by the DNN for the corresponding schedule. In [12], Wu et al. presented a deep reinforcement learning (DRL)-based scheduling strategy for allocating channels and time slots in the Industrial Internet of Things (IIoT). The proposed deep scheduling algorithm, which was named DISA, was intended to operate in an edge-computing-based network for communication scheduling.

This paper proposes an energy-efficient packet scheduling algorithm based on a DQN. We considered an application scenario of a WSN in which multiple slave nodes centered around the master node were connected in a star topology, similarly to [13]. The master collected the sensed data transmitted from the slaves while completely controlling the slaves’ transmissions through scheduling. In time-division multiple access (TDMA)-based transmissions with variable CIs [14], the master node controls the length of the CIs and the number of packets to be transmitted per CI for each slave. Furthermore, QoS requirements were imposed on packets, so each packet needed to be transmitted within a predetermined time. In this situation, it was not obvious to design an optimal scheduling algorithm that maximized the network’s lifetime while satisfying the QoS requirements. In particular, a traditional scheduling algorithm would hardly capture the complicated interplay between operating transmission parameters, the imposed constraints (e.g., QoS requirements, limited battery capacity, etc.), and the network lifetime.

To address the complexity of the problem, we relied on an AI technique called reinforcement learning (RL) [15]. In the framework of RL, the master incorporating the packet scheduler acts as an agent, and the rest of the parts of the WSN other than the master correspond to the environment. The master performs a sequence of scheduling actions over time and receives feedback in the form of state updates and rewards from the environment. Properly designed reward functions drive the DQN-based packet scheduler to converge to the desired form through learning over multiple episodes. Experiments were conducted in various network environments to measure the performance of the optimized DQN scheduler. In WSN environments, factors that affect the scheduler’s policy include the node population, mobility, data arrival process, packet lifetime, etc. Through extensive experiments, we examined how the DQN scheduler adapted to the time-varying network environment, and we compared the performance of the DQN scheduler with that of other existing methods in terms of the network lifetime and degree of QoS guarantee. Additionally, by analyzing the policy of the optimized DQN scheduler, we aimed to gain insight into scheduling methods based on RL. These efforts are expected to not only increase the understanding of the underlying processes of AI-based solutions, but also provide many insights into designing solutions for non-AI techniques through reverse engineering.

The rest of this paper is structured as follows. Section 2 presents the system model and the problem formulation. In Section 3, an energy-efficient packet scheduling algorithm based on a DQN is proposed. Section 4 presents the experimental methods and numerical results. In Section 5, a review of related work is presented. Finally, this paper concludes in Section 6.

## 2. System Model

### 2.1. Network Model

In this study, we consider a WSN model that consists of one master and multiple slaves connected to the master in a star topology, as shown in Figure 1a. All communication occurs between the master and one slave at a time, and the master switches between slaves via TDMA. A slave can transmit packets only when granted permission by the master. These characteristics of our model are similar to those of the BLE-connected mode in [13,14]. In fact, our network model can be viewed as an extension of the BLE-connected mode to accommodate additional functionalities required for AI-based scheduling.

After establishing the connection, the master and slaves communicate with each other during a predefined connection interval (CI), which consists of a control period (CP) and data period (DP), as shown in Figure 1b. In the CP, the master collects information on the transmission status from each slave. This information, including the number of packets in the queue and the packet delay, is used for scheduling decisions. In the DP, the scheduler determines the number of packets to be transmitted for each slave and grants permission to transmit packets by polling each slave. Since the CP is fixed and very small compared to the DP, the DP accounts for most of the CI. After the transmission is completed in the DP, the slaves return to the sleep mode until the CI ends, and they become active again when the next CI starts.

In this study, we assume that the master, which incorporates a packet scheduler, dynamically sets the CI length, as well as the number of packets that need to be sent during the CI for each slave. Such scheduling information is broadcast by the master, and the slaves then react accordingly upon receiving the information. The notion of QoS is incorporated into our system model, which assumes that the packets have a latency limit. For instance, if a packet in the queue waits longer than this latency limit, e.g., 10 s, the packet will be discarded by the slave and considered lost. Upon making a connection, the packet loss rate is specified as a QoS parameter between the master and slaves.

Regarding the energy model, we assume that the slave nodes are powered by a finite-capacity battery, although the master is supplied with sufficient energy via either a high-capacity battery or wired power sources [13,14]. As such, the network lifetime is mostly determined by the slave nodes, which become inoperable if the battery runs out. It is also assumed that energy is consumed by communication only, i.e., packet transmission or reception. One unit of energy is set to be consumed for the transmission/reception of a data packet that is 37 bytes long. For packets of other sizes, the energy consumption is proportional to the packet size.

### 2.2. Problem Formulation

To formulate the problem, we consider a WSN model that contains a master node and *N* slave nodes, as shown in Figure 1a. Since communication continues until the battery is exhausted, the lifetime of the *i*th slave, $V^{\left(i\right)}$, is defined as
$$V^{\left(i\right)}=\sum_{k=1}^{M}L_{k},\;\mathrm{for}\;e_{M}^{\left(i\right)}<E_{s}^{\left(i\right)}<e_{M+1}^{\left(i\right)}$$
where $L_{k}$ denotes the length of the *k*th CI, *M* represents the index of the last CI, $e_{k}^{\left(i\right)}$ denotes the cumulative power consumption of the *i*th slave until the *k*th CI, and $E_{s}^{\left(i\right)}$ represents the initial battery level of the *i*th slave. In this study, the network lifetime is defined as the time until one of the nodes runs out of battery for the first time. Assuming that the slaves use a limited amount of energy compared to the master, the network lifetime is completely determined by the slaves. Therefore, the slave node with the shortest lifetime determines the entire network lifetime.

We maximize the network lifetime by scheduling $\left(L_{k},{\left\{n_{k}^{\left(i\right)}\right\}}_{i=1}^{N}\right)$ for k=1,2,…, while ensuring the QoS requirements $q^{\left(i\right)}$ of each slave, i.e.,
(1)$$\begin{array}{cc} \underset{L_{k},{\left\{n_{k}^{\left(i\right)}\right\}}_{i=1}^{N}}{\mathrm{maximize}}\; & \min{\left\{V^{\left(i\right)}\right\}}_{i=1}^{N}\\ \mathrm{subject}\mathrm{to}\; & q^{\left(i\right)}>\eta^{\left(i\right)} \end{array}$$
where $q^{\left(i\right)}$ is the QoS metric of the packet transmission for the *i*th slave and $\eta^{\left(i\right)}$ is its target QoS requirement.

## 3. DQN-Based Scheduling Algorithm

We present a brief overview of reinforcement learning (RL) and discuss how the scheduling problems of the WSN can be formulated with the DQN.

### 3.1. Reinforcement Learning

RL is a class of machine learning (ML) approaches in which a decision-making agent learns the optimal policy in an environment by exploring various actions and observing the associated rewards [15,16]. Figure 2 shows the framework of RL [15], which is composed of an agent and the environment. At each time step, the agent observes the state of the environment and takes an action on the environment. The environment then transitions to a new state and responds to the agent by returning a reward. By repeating this process, the agent is able to learn a sequence of actions that maximize the cumulative reward without any a priori knowledge of the environment [15].

Q-learning, which considered one of the most popular RL algorithms [15], finds an optimal action policy by using a Q-function that represents the value of an action for a given state. The Q-function Q(s,a) at state *s* and action *a* is updated as follows:(2)$$\begin{array}{c} Q\left(s,a\right)\leftarrow Q\left(s,a\right)+\alpha [r+\gamma \max_{a}Q\left(s^{\prime },a\right)-Q\left(s,a\right)] \end{array}$$
where *r* denotes the reward given for taking an action *a*, α is the step size, γ is the discount factor, and $s^{\prime }$ is the next state [15]. In practice, the tabular update method in (Equation 2) suffers from explosive growth in the state–action (s,a) space and slow convergence for common real-world problems. To address these issues, the deep Q-network (DQN) proposed in [17] uses a neural network to approximate the Q-function. Moreover, two key components are incorporated into the DQN to stabilize the learning: experience replay and a duplicated Q-network [15,17]. Experience replay employs mini-batching to update the Q-network for better stability. The target network, which is duplicated from the main Q-network, is devised to improve the stability of the learning, and Q-values are used to train the main Q-network.

### 3.2. DQN-Based Scheduling Algorithm

The scheduling of transmission parameters in our WSN model is formulated as a sequential decision-making problem under the RL framework. In this formulation, the agent corresponds to the master node, and everything beyond the master node is considered the environment. The environment presents the system state and the reward to the agent, as shown in Figure 2. We assume that this state information is provided through the return packets from the slaves during the CP, as shown in Figure 1. The master node takes scheduling actions in response to the system state information by selecting appropriate transmission parameters.

Figure 3 shows the architecture of a DQN-based scheduling algorithm in the WSN environment that was described earlier. The master node observes a state $s_{t}$ at time *t* and takes an action $a_{t}$ by selecting the CI and the number of packets to transmit per CI. According to the master’s scheduling action, the communication between the master and the slaves takes place during the CI. At the end of the CI, a quadruple experience sequence $\left(s_{t},a_{t},r_{t},s_{t+1}\right)$, which represents the current observation state, current action, current reward at time *t*, and next state at t+1, is fed back to the master. These quadruple data are stored in a replay buffer, and a mini-batch is randomly sampled from the buffer to train the DNN. The main Q-network’s parameters are updated at each training step. Detailed specifications on the states, actions, and rewards are described below.

#### 3.2.1. States

The state of the environment $s_{t}$ at time *t* is represented by a vector of the remaining lifetime of the packets for the *N* slaves: (3)$$s_{t}=\left(D_{1},D_{2},\ldots ,D_{N}\right)$$
where $D_{i}=\left(d_{1},d_{2},\ldots ,d_{k}\right)$ indicates the remaining lifetime of the first *k* packets in the queue of the *i*th slave. The remaining lifetime of the packet is defined as the time left until the maximum latency since its arrival in the queue. A larger value of *k* will better represent the state, but will also cause an exponential growth in the state space S. Thus, we set *k* to 5 in our experiments without loss of generality. The remaining lifetime *d*, which is continuous in nature, was discretized into 20 levels for modeling purposes. Each level is 0.2 sec long, and this value was selected to accommodate the range of CIs (7.5 msec to 4 sec) specified in the BLE specifications [14].

#### 3.2.2. Actions

An action $a_{t}\in A$ at time *t* consists of two components: the CI length and the number of packets sent by each slave during the CI. Thus, the action $a_{t}$ is expressed as
(4)$$a_{t}=\left(L,n_{1},n_{2},\ldots ,n_{N}\right)$$
where *L* denotes the length of the CI and $n_{1},n_{2},\ldots ,n_{N}$ denote the numbers of packets that each *N* slave transmits during the CI. We set the value of $n_{i}$ to range from 0 to 5, considering that a maximum of five packets were observable for each slave node. According to the BLE specifications [14], the length of a CI in seconds, *L*, is given as
(5)L=0.0075+0.00125×i,i=0,1,…,3194,
where the index *i*, which is also called the CI index, usually refers to a specific CI. In our experiments, we used a CI index ranging from 0 to 31 instead of 0 to 3194 to avoid having a huge action space. These CI indexes covered from 7.5 msec up to 4 s with a step size that was 125 msec long.

#### 3.2.3. Rewards

The goal of our DQN-based packet scheduler is to maximize the network lifetime while satisfying the latency constraints. In the RL framework, this can be achieved by designing an appropriate reward function that is intended to guide the learning process. The reward function in our packet scheduler consists of two components: the CI length and QoS metric. The reward $r_{t}$ is formulated as
(6)$$r_{t}=\left(1+C_{id}\right)\times \prod_{i=1}^{N}q_{i}$$
where $C_{id}$ is the CI index in (Equation 5) at time *t* and $q_{i}$ is the QoS value of the *i*th slave at time *t*. $q_{i}$ is given by
$$q_{i}=\left\{\begin{array}{cc} 0, & \mathrm{if}\;n_{d}\neq 0\;\\ 1/2, & \mathrm{if}\;n_{i}=0,n_{d}=0\\ 1, & \mathrm{otherwise} \end{array}\right.$$
where $n_{d}$ denotes the number of dropped packets in the state $s_{t}$ for the *i*th slave. The reward function in (Equation 6) is designed to increase with a longer CI and fewer packet losses. The learning part of the DQN scheduling algorithm is presented in Algorithm 1.
**Algorithm 1:**DQN scheduling algorithm.  1:Initialize the replay buffer *B*;  2:Initialize the network *Q* with random weights θ;  3:Initialize the target network $\hat{Q}$ with random weights $\theta^{{}^{\prime }}=\theta$;  4:**for** each episode **do**  5:   **for** each step of the episode **do**  6:     With the parameter ϵ, select a random action *a*; otherwise, $a=arg\max_{a}Q\left(s,a\right)$;  7:     Take action *a*, observe *r*, $s^{{}^{\prime }}$, and store the transition tuple $\left(s,a,r,s^{{}^{\prime }}\right)$ in the replay buffer *B*;  8:     Sample a random mini-batch of transitions $\left(s,a,r,s^{{}^{\prime }}\right)$ from the replay buffer *B*;  9:     Set
$$y=\left\{\begin{array}{cc} r, & \mathrm{if}\mathrm{episode}\mathrm{ends}\\ r+\gamma \max_{a}\hat{Q}\left(s^{{}^{\prime }},a\right), & \mathrm{otherwise} \end{array}\right.$$ 10:     Perform a gradient descent step on loss function $L_{loss}={\left(Q\left(s,a\right)-y\right)}^{2}$ with respect to the network parameters θ; 11:     Every *K* steps, synchronize the network parameters $\theta^{{}^{\prime }}=\theta$; 12:   **end for** 13:**end for**

## 4. Numerical Results and Discussion

In this section, we analyze the performance of our DQN-based packet scheduling algorithm for a WSN model. The considered WSN model was composed of a master and multiple slaves connected to it in a star topology. The master scheduled the key operating parameters, including the CI and the number of packets to be transmitted for each CI. The DQN was used to optimize the scheduling algorithm that maximized the network lifetime while satisfying the QoS requirement. Hundreds of simulated episodes were used for training. An episode in our model refers to a communication session between the master and multiple slaves. The communication session was simulated by using a discrete-event simulation library named SimPy [18]. An episode started the communication session after initializing all of the relevant network parameters (e.g., arrival processes, packet queues, battery level, etc.) and ended when one of the slaves ran out of battery for the first time. Packet arrival processes for the slaves were assumed to follow a Poisson process with an arrival rate λ. Upon arrival, each packet was assigned a packet lifetime as a QoS parameter. The packet was to be dropped and treated as lost if the transmission was not completed within the packet lifetime. We assumed that energy was consumed only during communication, ignoring other energy uses. For the convenience of the experiments, 1.0 units of energy were consumed for transmission or reception of a data packet that was 37 bytes long. Packets of different sizes were assumed to consume energy proportional to this value. The battery level of the slaves was initially set to 1000.0 units, whereas the master’s was set to infinity. Due to this energy imbalance among the nodes, the network lifetime was entirely determined by the lifetimes of the slaves. The number of slave nodes was set to two unless otherwise noted. Our model was trained in the environment of OpenAI Gym [19], an open-source toolkit for developing RL applications. We developed our own WSN model and integrated it into OpenAI Gym. Some of the key parameters used in the experiments are listed in Table 1.

We evaluated the performance of our DQN-based scheduling algorithm in terms of convergence, cumulative rewards, network lifetime, and transmission ratio. Here, the transmission ratio is defined as the ratio of packets that were successfully transmitted from each slave node. For example, a transmission ratio of 0.95 means that 95% of the packets were successfully transmitted, but the rest were lost because they were not transmitted within the packet lifetime. Note that our DQN-based scheduling algorithm is hereafter abbreviated as DQN for convenience. The DQN was compared with a non-AI scheduling scheme, which is abbreviated as DET, in which the value of the CI was always chosen at its maximum, and as many packets as were in the queue were sent.

In Figure 4, we evaluate the performance of the DQN-based scheduler in terms of the cumulative reward, network lifetime, and transmission ratio by increasing the number of episodes to be learned. As shown in the figure, all three metrics converged while learning 350 episodes. In particular, the convergence became clear and steady after 100 or more episodes. According to the experimental results, the cumulative reward, network lifetime, and transmission ratio converged to 15,800, 1450, and 0.99, respectively. It was peculiar that a sharp increase in the network lifetime was observed around the 15th episode. This was caused by the action of ‘never transmitting’ becoming dominant during the early stage of learning. Note that our reward function in (Equation 6) was designed to favor a longer CI and no packet loss. Thus, a simple policy of ‘transmitting no packets, but taking the maximum CI’ can result in a relatively longer network lifetime. Obviously, this simple policy cannot be optimal, since the DQN-based scheduler maximizes the cumulative reward rather than the network lifetime. Consider that the value of the transmission ratio around the 15th episode is very low, whereas the corresponding network lifetime is large. Since the rewards are roughly a product of the network lifetime and transmission ratio, the value of cumulative rewards around the 15th episode remains almost unaffected. This suboptimal policy was overwhelmed by better policies and was eventually filtered out through learning.

Figure 5 shows the average cumulative reward, network lifetime, and transmission ratio for the DQN and DET. Here, the DET was divided into two cases, namely, DET2 and DET4, which denote the DET cases with packet lifetimes of 2 and 4 s, respectively. In the figure, we can see that performance of the DQN was similar to DET4’s, but was superior to DET2’s. DET4 could outperform DET2, since the larger packet lifetime of DET4 would reduce the packet loss rate for the DET schedulers, in which a fixed CI was used. Specifically, consider a packet that arrives during the CI. This packet will not be dropped due to its longer lifetime for the DET4 scheduler. However, the chance that this packet will be dropped becomes higher for the DET2 scheduler. Selecting a fixed CI length that is larger than the packet lifetime leads to higher packet losses and, consequently, aggravates the stability of the scheduling. DET4 accidentally operated with parameters that yielded optimal performance, but there was no guarantee of tuning to optimal parameters when operating in a time-varying communication environment. The results show that dynamic adjustment of transmission parameters through learning can show better performance than deterministic scheduling.

Table 2 lists the most frequent state–action pairs of the DQN scheduler under the default settings. After running the fully trained DQN scheduler for several episodes, these state–action pairs were collected. The table consists of state–action pairs and their frequencies. Since the state–action pairs governed the behavior of the DQN scheduler, they could show how the DQN scheduler acted at a particular state. As shown in the table, the all-zero state, which corresponded to the case of no packets in the queue, and the one-packet states accounted for most of the frequent (state, action) pairs. In general, how the group of frequent states was formed was greatly influenced by system parameters, especially traffic arrival rate. Two characteristic behaviors of the DQN scheduler can be observed in the table. First, the scheduler tended to select the almost-maximum CI, i.e., CI indexes of 29, 30, or 31. Note that 31 was the index of the maximum CI in our experiments. Second, packets were transmitted immediately, regardless of their remaining lifetimes. That is, the optimal DQN scheduler did not take actions such as intentionally deferring packet transmission or sending all packets in a batch. These findings are expected to be useful for reverse engineering AI schedulers or understanding the behavior of AI-based solutions.

## 5. Related Work

As one of the core technologies supporting the IoT, the WSN has attracted much attention among researchers. Most of the early relevant studies focused on designing different routing schemes and communication protocols to improve the energy efficiency of WSNs [20,21]. In [22], Qiu et al. proposed an urgency-based packet scheduling scheme named EARS. The master node determined the packet scheduling sequence while taking the packets’ priority and deadline into account. Packets with a higher priority were processed before the low-priority ones. If the priority was the same, the packet that expired sooner would be transmitted first. On the other hand, sleep scheduling mechanisms are widely used to prolong the network lifetime. In [23], Feng et al. proposed a scheduling strategy named EBMS for tracking targets in a cluster structure. It was intended to balance the energy in terms of multisensor distributed scheduling. The cluster header adaptively changed the sleeping time of cluster members based on the distance. To avoid missing the targets, cluster members closer to the cluster center would have a longer sleep time, while members closer to the cluster border would have a shorter one.

Inspired by the innovations brought by AI, several studies related to wireless networks have adopted AI-based techniques. In [9], Bhandari et al. proposed an SVM-based algorithm for proportional fair scheduling (PFS) of users on a channel. Compared to conventional metric-based PFS, which evaluates with a user-by-user metric, the SVM-based PFS algorithm used channel capacity and average throughput as the metrics for scheduling. In [10], Zhang et al. utilized a DNN to approximate the traditional SCA in channel assignment algorithms. The samples could be used to train a user’s channel assignment policy quickly in a real-time environment. The simulation results showed that the proposed algorithm significantly reduced the computation time and algorithm complexity. In [11], Xu et al. proposed DQN-based link scheduling for multiple small base stations. The scheduler selected a link based on channel gains and transmission weights. The simulation results showed that DQN-based scheduling required less computation while achieving similar performance to that of an exhaustive search. In [12], Wu et al. presented a DRL-based scheduling strategy for allocating channels and time slots in an IIOT. The scheduling algorithm considered various network information, including the source node, target node, number of transmission time slots, and time slot occupancy. Their scheduling algorithm was reported to have a higher success rate and schedulability compared with those of other alternative methods.

BLE, which provides low-power and short-range wireless connectivity, has been considered as a communication technology that is suitable for IoT. Thus, many studies have investigated the enhancement of the performance of BLE-based communications in the IoT [24,25,26,27,28]. In [25], Shan et al. introduced a method for reducing the detection time of the surrounding advertisers by tuning the advertisement interval. By doing so, their method could reduce unnecessary energy consumption. In [26], Ghamari et al. proposed a packet collision model to estimate an advertisement collision when multiple nodes transmitted advertisement packets simultaneously. The experimental results showed that decreasing the advertisement interval significantly increased the possibility of packet collisions, thus increasing the energy consumption of the nodes. In [27], Giovanelli et al. evaluated three BLE modules in terms of throughput, latency, and power consumption. The results showed that optimizing the connection parameters, such as the CI, could improve the system efficiency while maintaining the required throughput. In [28], Fu et al. proposed Q-learning-based resource scheduling in a WSN to enhance the energy efficiency while providing a QoS guarantee specified by a packet loss rate.

To the best of our knowledge, most existing IoT-related studies considered energy efficiency and throughput as performance metrics; however, only a little work considering the QoS has been reported. Nonetheless, the studies by Rioual et al. in [29] and Collotta et al. in [30] are relatively close to our research. In [29], Rioual et al. used Q-learning to manage the energy of sensor nodes by adjusting the sleep duration. This scheme tuned the processor frequency of the nodes by considering the energy harvested from the environment. In particular, they found that Q-learning is sufficient to handle a moderate size of this problem, though it is less appropriate for a large or nearly infinite state space. In [30], Collotta et al. proposed a fuzzy-based scheme that scheduled the sleeping time of IoT nodes by considering the battery levels and the throughput-to-workload ratio. Compared to the schemes that used a fixed sleeping time, the proposed method increased the node lifetime by 30%. However, this simulation was based on a simple network environment (i.e., only one peripheral device), and the sleeping duration was ambiguous due to the inclusion of unknown environmental parameters.

## 6. Conclusions

In this study, we proposed a practical DQN-based packet scheduling algorithm that coordinates the transmission of multiple slaves in a WSN. The scheduler embedded in the master dynamically adjusts the CI and the number of packets transmitted by each slave within the interval. The experimental results confirmed that the DQN scheduler can adapt to dynamic network environments via continuous learning and can prolong network lifetime while providing QoS guarantees. Moreover, an in-depth analysis of the optimized scheduler’s policy revealed that immediate transmission of packets using the maximum CI is optimal, rather than using batch transmission by deferring the transmissions. As future research suggestions, there may be a need for ways to reuse previously learned policies in highly mobile network environments in which the node population in the WSN varies significantly over time. It is also interesting to analyze how the optimal policy changes as the amount and pattern of scheduling overhead vary.

## Figures and Tables

**Figure 1 sensors-23-01339-f001:**
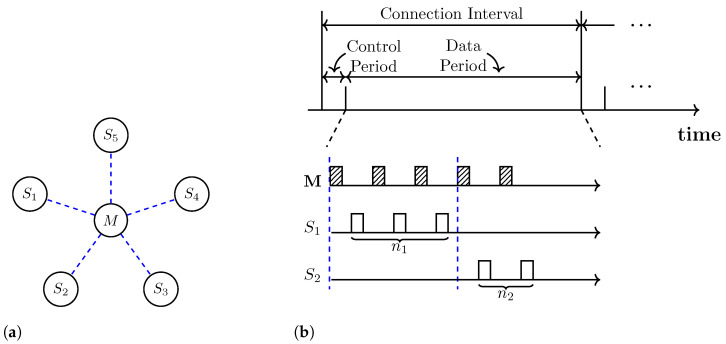
(**a**) Master *M* and multiple slaves $S_{i}$ connected in a star topology. (**b**) Packet exchanges between the master and slaves over time. The $n_{i}$ denotes the number of packets trasnsmitted by the *i*th slave.

**Figure 2 sensors-23-01339-f002:**
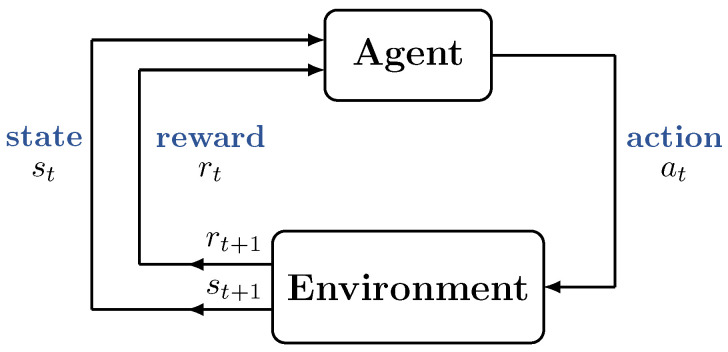
The interaction between an agent and the environment in the framework of reinforcement learning [15]. At a time step *t*, the agent observes the state $s_{t}$ of the environment and takes an action $a_{t}$. The environment then transitions to a new state $s_{t+1}$ and responds to the agent by returning a reward $r_{t}$.

**Figure 3 sensors-23-01339-f003:**
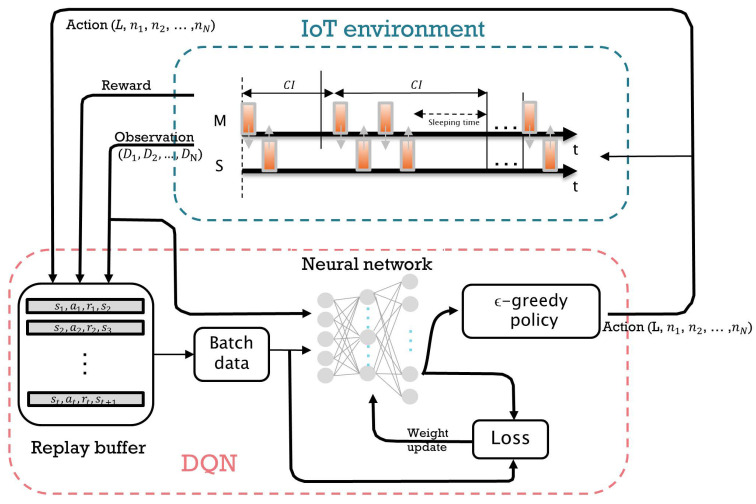
Architecture of the DQN-based scheduling algorithm. At the beginning of the CI, the DQN-based scheduler receives state information (i.e., the remaining lifetime of packets in the queue $\left(D_{1},D_{2},\ldots ,D_{N}\right)$) from the slaves. Then it performs scheduling actions, i.e., choosing the CI length and the number of packets transmitted by each slave $\left(L,n_{1},n_{2},\ldots ,n_{N}\right)$. The reward *r* is given to the scheduler as feedback along with the updated state $s_{2}$. Such a sequence of relevant information (i.e., current state $s_{1}$, action *a*, reward *r*, and next state $s_{2}$) is stored in the replay buffer for the training of the DQN-based scheduler.

**Figure 4 sensors-23-01339-f004:**
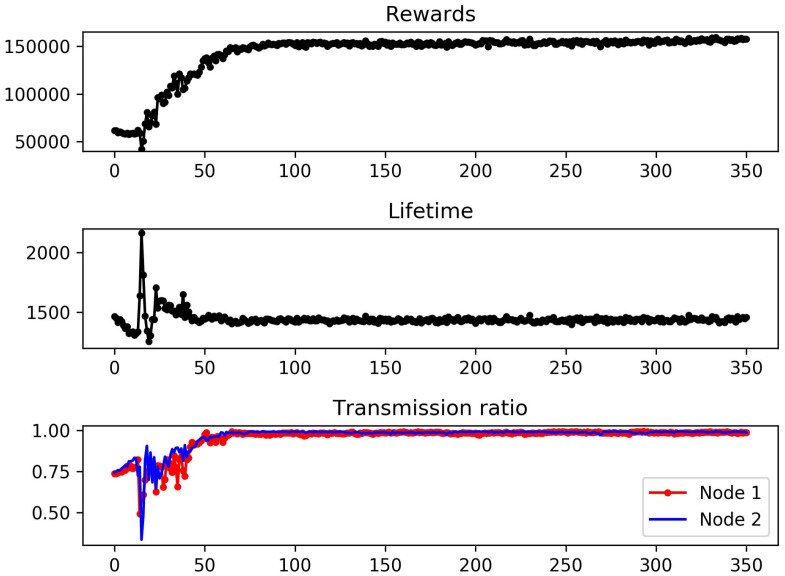
(**Top**) Cumulative rewards versus the number of episodes for the DQN. (**Middle**) Network lifetime versus the number of episodes for the DQN. (**Bottom**) Transmission ratios versus the number of episodes for the DQN.

**Figure 5 sensors-23-01339-f005:**
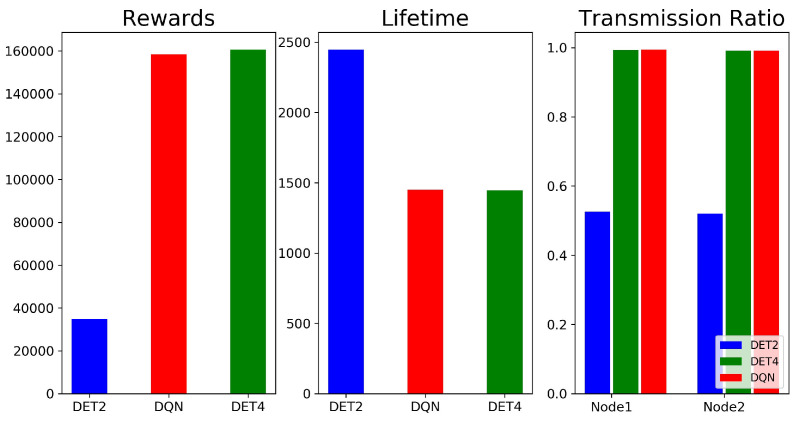
Average cumulative reward, network lifetime, and transmission ratio for the DQN and DET. DET2 and DET4 denote the DET cases with packet lifetimes of 2 and 4 s, respectively.

**Table 1 sensors-23-01339-t001:** Parameters used in the numerical experiments.

Parameters	Values
Connection interval length (*L*)	0.0075∼3.8825 s
Discretized delay levels	20
Packet arrival rate (λ)	0.5/s
Transmission speed of link	1 Mbps
Interframe space (IFS)	0.15 ms
Master packet size	12 bytes
Slave packet size	37 bytes
Packet lifetime (τ)	2–4 s
Initial battery capacity	1000.0
Energy consumed by transmitting/receiving a data packet	1.0
Energy consumed by transmitting/receiving an empty/control packet	0.3243

**Table 2 sensors-23-01339-t002:** The most frequent (state, action) pairs for the DQN scheduler.

States	Action	Frequency
((0,0,0,0,0), (0,0,0,0,0))	(29,0,0)	3.275
((0,0,0,0,0), (16,0,0,0,0))	(30,0,1)	0.425
((19,0,0,0,0), (0,0,0,0,0))	(31,1,0)	0.35
((16,0,0,0,0), (0,0,0,0,0))	(31,1,0)	0.35
((10,0,0,0,0), (0,0,0,0,0))	(31,1,0)	0.35
((0,0,0,0,0), (11,0,0,0,0))	(30,0,1)	0.325
((17,0,0,0,0), (0,0,0,0,0))	(31,1,0)	0.325
((15,0,0,0,0), (0,0,0,0,0))	(31,1,0)	0.325
((0,0,0,0,0), (4,0,0,0,0))	(30,0,1)	0.325
((20,0,0,0,0), (0,0,0,0,0))	(31,1,0)	0.3

## Data Availability

All data derived from this study are presented in the article.

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
