# Peer review of "Deep-Q-Network-Based Packet Scheduling in an IoT Environment"

_sensors, 2023, doi:10.3390/s23031339_

Round 1
Reviewer 1 Report
The manuscript “Deep Q-Network Based Packet Scheduling in IoT Environment” investigates a practical Deep Q-Network (DQN)-based packet scheduling algorithm that coordinates the transmission of multiple IoT devices. The manuscript needs the following improvement to enhance the quality of the manuscript:
· In Table 1, mention the unit of connection interval length, Initial battery capacity and Energy consumed by transmitting/receiving data packet.
· The authors also mention the reason for the low transmission ratio when the value of the network lifetime around the 15th episode was large.
· The calculation needs to be revised because it is the repetition of the abstract. In conclusion, there should be a summary of the main points of the study.
· The grammar of the manuscript also needs to be revised.
Author Response
Dear reviewer,
Thank you for giving me the opportunity to submit a revised draft of my manuscript titled "Deep Q-Network Based Packet Scheduling in IoT Environment" to MDPI Sensors. We appreciate the time and effort that you have dedicated to providing your valuable feedback on my manuscript. We are grateful to the reviewers for their insightful comments on my paper. We have been able to incorporate changes to reflect most of the suggestions provided by the reviewers. Please see the attachment for detailed responses.

Reviewer 2 Report
This paper proposes a practical packet scheduling algorithm based on deep Q network (DQN), which coordinates the transmission of multiple IoT devices. The problems solved by this solution are very interesting, but there are still some problems that need to be explained and improved:
1. In the Abstract, the authors should introduce more about the work of this article.
2. In the introduction, the motivation of research should be further emphasized.
2. What are the advantages of packet scheduling algorithm based on deep Q network? Compared with other algorithms, which performance indicators are superior? Are there quantitative indicators? Can these quantitative indicators be given in the conclusion part?
3. What is the role of "Related Work" in Section 5? Compare? Or prospect?
4. Some numbers are too vague to be recognized. As shown in Figure 3.
5. The descriptions of some statements are incorrect and need to be carefully modified. For example, "The amount of energy consumed by transmitting or receiving variant packet types is listed in Table 1." Table 1 in the text seems to be inconsistent with this description. The colon "is updated as follows:" needs to be deleted, which is consistent with other parts of the text.
Author Response

(The authors gave the same response as above.)

Round 2
Reviewer 1 Report
The authors submitted the revised manuscript, “Deep Q-Network Based Packet Scheduling in IoT Environment”. The authors responded according to comments provided by the reviewers. In present form manuscript can be considered for the publication.
Reviewer 2 Report
The modified version can be accepted.